# Evaluation of the Potential Flight Ability of the Casuarina Moth, *Lymantria xylina* (Lepidoptera: Erebidae)

**DOI:** 10.3390/insects15070506

**Published:** 2024-07-06

**Authors:** Jifeng Zhang, Baode Wang, Haojie Ren, Jianing Chen, Junnan Li, Yuanyuan Sun, Yonghong Cui, Rong Wang, Mengxia Liu, Feiping Zhang

**Affiliations:** 1College of Forestry, Fujian Agriculture and Forestry University, Fuzhou 350002, China; zhangjif118@163.com (J.Z.);; 2US Department of Agriculture, Animal and Plant Health Inspection Service, Forest Pest Methods Laboratory, Riverdale, MA 02542, USA; 3Fujian Academy of Forestry Sciences, Fuzhou 350012, China

**Keywords:** *Lymantria xylina*, flight ability, flight mill, dispersal, female moth

## Abstract

**Simple Summary:**

The casuarina moth, *Lymantria xylina* Swinhoe (Lepidoptera: Erebidae), may become a major invasive pest similar to *Lymantria dispar* Linnaeus (Lepidoptera: Erebidae), and the risk of its spread has attracted worldwide attention. Male *L. xylina* moths have strong flight ability, while female moths have weaker flight ability. Female moths frequently rest on branches and can sometimes fly short distances. Additionally, females may be attracted by the lights of ships and land on the hull or cargo to lay eggs. Prior to this study, little was known about the flight ability of *L. xylina*, due to the absence of related research. Consequently, our investigation focused on assessing the potential flight ability of *L. xylina* adults using a flight mill. We conducted a comprehensive evaluation of the flight ability of *L. xylina* moths in terms of age, sex, flight duration, mating, oviposition, wing morphology, and under different temperatures. The goal was to provide a theoretical framework for understanding the flight characteristics and dispersal potential of *L. xylina*, as well as to assess the risk of dispersal through ocean-going freighters.

**Abstract:**

*Lymantria xylina* Swinhoe (Lepidoptera: Erebidae) is a potentially invasive pest, similar to *Lymantria dispar asiatica* Vnukovskij and *Lymantria dispar japonica* Motschulsky (Lepidoptera: Erebidae). To evaluate its potential for spread and flight distance related to egg deposition on vessels at ports, we employed a flight mill to assess the flight capabilities of its adults under varying conditions. Our findings revealed that females primarily flew short distances and ceased flying after 3:00 AM, whereas males covered much longer distances throughout the day. Sex, age, and flight duration significantly influenced flight ability. Females exhibited weaker flight capability than males, and their ability declined with increasing age or flight duration. Notably, 1-day-old moths displayed the strongest flight ability, with average flight distances of up to 3.975 km for females and 8.441 km for males. By the fifth day, females no longer flew, and males experienced reduced flight ability. After continuous hanging for 16 h, females lost most of their flight capacity, while males remained capable of flight even after 32 h. Additionally, female flight ability decreased significantly after mating, possibly due to factors such as egg-carrying capacity, weight, and load ratio. This study provides a foundation for assessing the risk of long-distance dispersal of *L. xylina* via ocean-going freighters, considering female moths’ phototactic flight and oviposition.

## 1. Introduction

Research on insect dispersal is very important for controlling the spread of insects [1,2]. Insects’ flight behavior is important for understanding insect dispersal. Research on insect flight behavior includes two aspects: the study of flight patterns, and the study of flight ability [3]. Primary methods for studying insect flight patterns include ovary dissection, field observation, mark–release–capture, and trapping [4,5,6,7]. In recent years, many new technologies have been applied to the study of insect flight characteristics, such as three-dimensional (3D) reconstruction technology and individual tracking observation [8,9]. The flight ability of insects plays a decisive role in their natural dispersal paths and range [10]. The methods for studying insects’ flight ability include the measurement of lifting force, wingbeat frequency, wing loading, three-dimensional space video tracking (Track 3D), the use of flight mills, etc. [11,12,13,14].

Flight mill technology is widely used to study the flight patterns and flight ability of insects [15], and to reveal the influences of environmental factors, food nutrition, mating, and oviposition on the flight ability of insects; this approach also plays an important role in the study of interactions between insects’ flight ability and reproduction and their physiological and metabolic changes during flight [16,17]. Flight mill technology is simple and easy to use and provides an intuitive and reliable basis for determining the flight ability of insects. However, the test insects are forced to fly and, thus, may exhibit extraordinary flight ability under these restrained flight conditions due to escape responses. Therefore, researchers generally refer to flight abilities determined using a flight mill as potential flight abilities [18,19].

Sex, age, mating status, temperature, and wing morphology are all important factors affecting the flight ability of insects [20,21]. Yang et al. [22] compared the flight ability of *Lymantria dispar* Linnaeus adults of varying age, sex, and reproductive status using a flight mill. The results showed that males have stronger flight ability than females, and that females’ flight ability decreases with increasing age as well as after oviposition. In addition, a cluster analysis was performed on the flight ability of *L. dispar* females from different geographic populations, showing significant differences in their flight abilities. Shi et al. [23] conducted a wing morphometry analysis on eight geographic populations of *L. dispar* adults, including indicators such as weight, wing length, wing width, wing area, aspect ratio, and load ratio, and they found that different geographic populations of *L. dispar* adults have different morphological characteristics and that their flight abilities show polymorphism.

In China, *Lymantria xylina* Swinhoe is currently found only in the provinces of Zhejiang, Fujian, Guangdong, Guangxi, and Taiwan. It poses a significant threat due to its vast host range. It can damage over 424 plant species from 103 families (including three subfamilies), with 58 being economically important crops. The range of plants that the larvae feed on expands with the increase in insect age, and the host species and damage areas are increasing year by year. *L. xylina* has become one of the main pests of shelter forests and economic fruit trees in southeastern coastal areas of China [24,25,26,27]. Due to its taxonomic proximity to *L. dispar* and similar biological characteristics, *L. xylina* has also been considered a major invasive pest, gaining international attention and being considered to pose a potential risk of biological invasion [27,28]. Females, drawn to light sources in port areas, lay eggs on shipping containers and vessels. If these contaminated vessels reach suitable climates and the eggs hatch after a dormant period (diapause), *L. xylina* could establish itself in new areas. Studies suggest *L. xylina* has the potential to become a serious invasive pest in warm US regions like Hawaii, Southern California, and Florida [28]. The highly damaging *Lymantria dispar dispar* Linnaeus highlights the challenges of invasive species control. After colonizing the Eastern US, *L. d. dispar* became a major forestry threat. Despite spending around USD 860 million annually on control efforts, the US government has been unable to eradicate the species [29]. This experience underscores the importance of preventing introductions. In recent years, a few countries, such as the US and Canada, have implemented stricter pathway risk mitigation measures on *Lymantria dispar asiatica* Vnukovskij and related species, aiming to prevent their arrival and the disruption that they could cause to international trade [28,29].

*Lymantria xylina* can disperse actively through the ballooning of early-instar larvae, crawling by older larvae, and adults flying, as well as passively with the movement of egg masses on substrates such as the surfaces of containers or vessels through trade, transportation, tourism, and other activities [26,28]. Female *L. xylina* moths exhibit a variety of behaviors, including resting on branches, crawling, and short flights [29,30,31]. They may also be attracted to artificial lights, potentially leading them to ships, where they could lay eggs on the hull or cargo [28,31]. However, conflicting reports exist regarding their overall flight ability. Some studies suggest that females can fly at night, while others describe them as having weak flight due to their size [30,32]. Additionally, the capture of a large number of females using a high-pressure mercury lamp [33] suggests some phototactic flight, while the discovery of female adults on ships entering ports suggests the existence of long-distance flight capability [34].

To resolve this ambiguity regarding flight ability, we employed a flight mill to assess the flight capacity of *L. xylina* adults. We conducted an investigation into how factors such as age, sex, mating status, egg-laying behavior, and temperature impact the flight performance of these moths. The wing morphology of newly emerged adults (1-day-old) was also compared between sexes.

To thoroughly evaluate the flight potential of *L. xylina* adults and understand their dispersal patterns, we employed a flight mill to assess their flight capacity across various factors. These factors included adult age, sex, mating status, egg-laying (oviposition), and temperature. Additionally, the wing morphology of newly emerged (1-day-old) adults was compared between sexes. This comprehensive approach provides a theoretical foundation for understanding their flight behavior and dispersal abilities. Ultimately, this knowledge is crucial for assessing the risk of long-distance spread via cargo ships.

## 2. Materials and Methods

### 2.1. Sources of Insects 

*Lymantria xylina* pupae were collected from Pingtan County, Fujian Province, China (119°45′08.55″ E, 25°25′13.28″ N) in early June 2022 and placed individually in 200 mL transparent plastic boxes (25 °C ± 1 °C, 40–50% RH, and L16: D8). Adults were weighed after eclosion, and the sex, date, and serial number were recorded for future use.

### 2.2. Instruments

The flight ability test was performed using a flight mill (FXMD-24-USB, Jiaduo Scientific, Industry, and Trade Co., Ltd., Hebi, China). This system can automatically record changes in the flight speed, total flight duration, total flight distance, number of flights, grounding time, distance per flight, and duration per flight of insects during flight, as well as performing preliminary analysis and processing of test data. The rotating arm of the flight mill is a stainless steel wire of 30 cm in length and 0.4 mm in diameter. The sampling error is less than one turn.

### 2.3. Experimental Design

We considered six factors in this experiment: sex, age, flight duration, temperature, mating status, and oviposition status. Individual adult moths were tested only once. The specific test settings were as follows:Different ages and sexes: Adults of different ages (1, 2, 3, 4, and 5 days old) and sexes (male and female) were collected twice daily, at 8:00 and 20:00. The number of moths tested varied by age and sex. The breakdown was as follows: females (7, 7, 6, 5, and 4, respectively); males (7, 6, 5, 5, and 6, respectively).Different flight durations: One-day-old female moths continuously flew for 24 h, and male moths continuously flew for 32 h. The flight data were collected every 8 h. Five female moths and six male moths participated in this experiment.Different temperatures: One-day-old male moths continuously flew for 8 h at different temperatures (22 °C, 25 °C, and 28 °C). The indoor temperature was controlled by air conditioning. The number of male moths in each treatment was 5, 4, and 3, respectively.Different mating statuses: Three pairs of 2-day-old female moths and 3-day-old male moths were placed in clear glass containers (measuring 62 cm × 41 cm × 48 cm) along with *Casuarina equisetifolia* branches. Their mating behaviors were observed hourly. Upon confirmation of successful mating, the adults were immediately transported (a 1.5-h drive) to the flight mill lab for testing.Different days after ovipositing: Pairs of male and female moths were placed in clear glass containers with *C. equisetifolia* branches, and egg-laying was observed once an hour. The time when the female moths started to lay eggs was recorded, and the number of days after ovipositing was calculated from the start of oviposition. The female moths were tested 1, 2, 3, or 4 days after ovipositing. The number of females tested at each timepoint was 5, 4, 5, and 3, respectively.

### 2.4. Test Methods

The flight tests were conducted in a near-natural light environment [35,36]. The flight mill was positioned near a north-facing window in a room free from artificial lighting or shading devices. Sensor data from the flight mills confirmed an illumination range of 0–0.137 lux (0.012 lux in average) during testing. Temperature and humidity were set to mimic typical conditions within *C. equisetifolia* forests during this period. Humidity was maintained at 70% ± 5% RH, while the temperature was held constant at 25 °C ± 1 °C, except for specific trials conducted at 22 °C and 28 °C. The flight tests were performed as described by Yang et al. [22], with some adjustments, usually starting at 9:00 AM and lasting 24 h. The flight test and data collection procedures were as follows:Anesthesia: Only *L. xylina* adults with fully developed wings and normal flapping ability after eclosion were selected for the experiments. To anesthetize the adult individuals, 7 drops of 50% ethyl acetate were injected into a cotton ball within the insect box. This rendered the adults temporarily unable to flap their wings but able to twitch slightly, allowing us to proceed with the experiment.Weight measurement: The anesthetized insects were weighed on an electronic scale (OHAUS, AR2140, New Jersey, NJ, USA).Fixation: To prepare for testing, the hair and scales on the pronotum of the test insect were carefully removed. A small, homemade plastic tube (2.2 mm in diameter, 2.0 mm in height) was then attached to the pronotum using clear, quick-drying glue. The glue was allowed to set for a few seconds.Flight test: The other end of the copper wire (0.1 mm in diameter) attached to the plastic tube was secured to the rotating arm of the flight mill. The ground base of the flight mill stood 2.0 cm tall. The test insect was then allowed to rotate freely tangentially around the base, with its flight direction aligned tangentially to the flight trajectory.Flight data acquisition: The insect flight information system was activated, and the parameters were set, including a rotating arm radius of 15 cm. The test insects naturally regained consciousness, and the flight mill automatically recorded the flight data. At the conclusion of the flight test, the test insects, along with the copper wire and small plastic tube, were removed. Subsequently, the copper wire and plastic tube were detached from the insects.Measurement of morphological indicators: One wing was carefully removed from its base and spread flat on graph paper with a 1 mm grid for measurement. A glass slide was used to gently press the wing flat. The moth’s body was then positioned flat with the pronotum (upper back plate), the periproct, and the bases of both wings exposed, as shown in Figure 1. Using digital calipers, the wing length, wing width, body length, and other relevant morphological features were measured. In addition to linear measurements, wing area was determined using digital image analysis. A Canon EOS 600D DSLR camera with an EF-S18-200 mm lens was used to capture images of the moths. These images were imported into Capture 2.3.1 software, typically used with an electron stereomicroscope (Nikon SMZ800N in this case). Within the image, a reference area of 1 cm^2^ (10 mm × 10 mm) was established using the scale bar. The forewing and hindwing areas were then measured relative to this reference area to calculate their actual sizes.

Table 1 lists the morphological parameters of an *L. xylina* adult. The abdominal width is the width of the widest part of the abdomen (the first abdominal segment is the widest in male moths, and the fifth abdominal segment is the widest in female moths). The thorax width is the width of the mesothorax. The body length is the length from the calvaria to the periproct. The abdominal length is the length from the constriction between the thorax and abdomen to the periproct.

Egg masses and individual eggs laid by female moths were collected and counted. To determine the total egg-laying capacity, the number of eggs remaining in the abdomens of dissected females was added to the number of laid eggs.

Finally, the total flight distance of one-day-old male and female moths was categorized into three groups (<500 m, 500–5000 m, and >5000 m). These flight distance categories served as thresholds to analyze the relationships between various morphological features (wing length, wing area, etc.) and the potential flight ability of *L. xylina* adults.

### 2.5. Data Analysis

The software MDB Viewer Plus (Microsoft Corporation, Albuquerque, NY, USA) was used to open preliminary analysis reports and detailed flight data for each test insect. The flight speed (km/h), total flight distance (km), total flight duration (h), and number of flights were used as primary indicators, and the duration per flight (h), distance per flight (km), and maximum uninterrupted flight distance (km) were used as secondary indicators; with these, the flight ability of *L. xylina* adults was evaluated.

Data analysis was performed using SPSS 22.0 software (IBM). Data were expressed as the mean ± standard error. If the data did not meet the assumptions of normality and homogeneity of variance, a logarithmic transformation was applied before further statistical analysis. For comparisons between two groups, the independent-samples *t*-test was used. For comparisons involving more than two groups, analysis of variance (ANOVA) was followed by Duncan’s multiple range test to identify significant differences between specific groups.

## 3. Results

### 3.1. Potential Flight Ability of L. xylina Adults

#### 3.1.1. Frequency Distribution of the Maximum Uninterrupted Flight Distance

Similar to the method used by Yang et al. [22] to define maximum uninterrupted flight distance in *L. dispar*, a threshold of 500 m was used to categorize flights of *L. xylina* adults as long-distance (greater than 500 m) or short-distance (less than 500 m) based on flight mill measurements (Figure 2a). The results showed a clear difference between male and female flight behavior. A significant majority (86.8%) of female moths engaged in short-distance flights. In contrast, male moths exhibited a more balanced distribution, with nearly half (54.9%) undertaking short-distance flights and the remaining individuals flying longer distances. The averages of all distances, short-distance, and long-distance among female moths were 0.651 km, 0.116 km, and 4.163 km, respectively, while those for male moths were 1.764 km, 0.155 km, and 3.886 km, respectively.

#### 3.1.2. Frequency Distribution of Total Flight Distance

As shown in Figure 2b, the total flight distances of *L. xylina* adults followed a similar pattern to the maximum uninterrupted flight distances. A greater proportion of females (52.8%) exhibited short-distance flights (under 500 m). In contrast, males displayed a clear dominance in long-distance flights (78.4%), with only 21.6% undertaking short-distance flights. This distribution suggests a generally weaker flight capability in female *L. xylina* moths compared to males.

### 3.2. Effects of Age and Sex on Flight Ability

The flight mill tests revealed significant effects of both sex and age on the flight capability of *L. xylina* adults (Figure 3). Flight ability generally declined with increasing age for both sexes (Figure 3b,c). Statistical analysis confirmed this trend, with strong correlations observed between age and flight duration/distance for both males and females: for the total flight duration of the male, *r*^2^ = 0.815, *p =* 0.036; for the total flight duration of the female, *r*^2^ = 0.840, *p* = 0.029; for the total flight distance of the male, *r*^2^ = 0.9525, *p* = 0.004; and for the total flight distance of the female, *r*^2^ = 0.845, *p =* 0.027. One-day-old adults exhibited the strongest flight, with average flight distances of 3.975 km and 8.441 km for females and males, respectively (Figure 3c). Interestingly, no significant difference in flight speed was detected between male and female moths of the same age (Figure 3a). The number of flights displayed greater variability, with females generally undertaking fewer flights (Figure 3f).

Female moths experienced a particularly rapid decline in flight performance after reaching 2 days of age. Their flight duration per flight, distance per flight, and total flight distance dropped significantly compared to one-day-old females (Figure 3c–e). By 4 days old, most females had lost their ability to fly altogether. While males also exhibited a decrease in flight ability with age, their decline was more gradual. They retained some flight capability even at 5 days old (Figure 3).

### 3.3. Circadian Rhythms of the Flight Behavior of L. xylina Males and Females

To investigate how flight behavior varies throughout the day (circadian rhythm), one-day-old adults were chosen due to their peak flight ability. Flight duration and the number of flights for both male and female *L. xylina* adults were recorded on the flight mill every hour (Figure 4). Female moths exhibited a clear preference for daytime flight on the flight mill. Their peak flight duration and number of flights occurred between 11:00 AM and 3:00 PM. A small number of flights were observed between 9:00 PM and 3:00 AM, but no flight activity was recorded beyond that time. In contrast to females, male moths displayed flight activity throughout the day. While their flight duration and number of flights showed some variation, with a general increase between 1:00 PM and 4:00 PM, there was no distinct overall pattern.

### 3.4. Impact of Flight Duration on Flight Ability

Figure 5 explores how flight duration affects flight performance in *L. xylina* adults. The average flight distance, flight duration, distance per flight, and duration per flight were measured every 8 h. Female moths exhibited a clear decline in flight performance as the flight duration increased. Their flight indicators (distance, total duration, distance per flight, and duration per flight) all showed a significant decrease. Flight ability was largely lost after 16 h of continuous flight, and some females even died after 24 h. In contrast to females, male moths displayed no significant changes in their flight indicators throughout the entire testing period. They maintained a certain degree of flight capability even after 32 h of continuous flight.

### 3.5. Effects of Different Ambient Temperatures on Flight Ability

The effects of different ambient temperatures (22 °C, 25 °C, and 28 °C) on the flight ability of male *L. xylina* adults were investigated. Appendix A (see Appendix A) shows the flight parameters measured at these temperatures. No significant differences were detected in any of the flight parameters (presumably distance, duration, speed, etc.) between the different temperature groups. This suggests that male *L. xylina* moths can fly normally within this range of ambient temperatures.

### 3.6. Effects of Mating Status on Flight Ability

Table 2 summarizes the flight parameters of *L. xylina* adults based on their mating status. Interestingly, mating appears to have a minimal impact on the overall flight ability of male moths. No significant differences were detected in flight distance, duration, speed, flight duration per flight, or distance per flight between mated and unmated males. However, mated males undertook a significantly higher number of flights compared to unmated males. In contrast to males, mating significantly affected the flight ability of female moths. Mated females exhibited a lower total flight duration and number of flights compared to unmated females. Additionally, their total flight distance was also significantly shorter, suggesting a decrease in overall flight capability after mating.

### 3.7. Impact of Oviposition on Flight Ability

The effect of oviposition on the flight ability of female *L. xylina* adults was examined. During the oviposition process on the flight mill, females exhibited various behaviors, including stopping to lay eggs, laying eggs while struggling, laying eggs in flight, and laying eggs between flights.

Figure 6 illustrates the flight parameters of female moths at different days following oviposition. The results show a clear negative impact of oviposition days on flight ability. Several flight parameters were significantly affected as the number of days since oviposition increased: total flight duration, total flight distance, and number of flights. All of these indicators displayed a consistent trend, decreasing progressively with each passing day after oviposition. This decline suggests a substantial reduction in female flight capability over time following egg-laying.

### 3.8. Relationship between Morphology and Flight Ability

Table 3 and Appendix A present the morphological characteristics of one-day-old *L. xylina* adults categorized by their total flight distances. No significant differences were detected in any of the 15 morphological measurements (listed in the tables) between male moths with varying flight distances. This suggests that body morphology is not a major factor influencing flight capability in male *L. xylina*. In contrast to males, female adults exhibited a clear link between morphology and flight ability. Significant differences were found in weight (*F* = 9.312, *df* = 2, *p* = 0.003) and load ratio (*F* = 7.827, *df* = 2, *p* = 0.005) based on their total flight distances. Interestingly, females with the shortest flight distances carried the greatest weight and had the highest load ratio (weight relative to wing area). For instance, the number of eggs carried by females in the shortest flight distance group (under 500 m) was significantly higher (604.7 ± 150.1) compared to females in the longest flight distance group (over 5000 m, 350.7 ± 66.9). This trend suggests that female flight ability may be influenced by the number of eggs they carry, their overall body weight, and the resulting load ratio.

## 4. Discussion

Flight Performance and Its Ecological Implications: The flight mill assay revealed significant effects of sex, age, and flight duration on the flight ability of adult *L. xylina* moths (Figure 3, Figure 4 and Figure 5). These findings align with field observations, where females exhibited limited flight activity during the night (21:00–3:00), followed by complete cessation, while males demonstrated long-distance flight capabilities throughout the day and night.

Sex-Based Differences and Reproductive Investment: Flight patterns and capabilities differed significantly between sexes. Females primarily undertook short flights with minimal activity, likely due to their focus on reproduction. Following emergence, they remained stationary or exhibited short-range movements while emitting sex pheromones to attract mates. Once mated, they laid eggs within a short timeframe (20 min to 17 h) and typically ceased flight entirely afterward [29,30,31,37]. This behavior aligns with their larger body size and weight compared to males, resulting in a higher load ratio, which hinders flight (Table 3). Similar observations were reported for *Hemileuca maia* Drury, where egg-laden females exhibited weak, short flights [38,39,40]. Field observations in China further supported this, revealing a prolonged pre-flight warm-up period and short flight distances in females, with egg-laying females lost their flight capability entirely (unpublished data). In fact, female *L. d. dispar*, which are much heavier and larger than the males, are reportedly losing their flight capability entirely, to stress that the *L. d. dispar* female is flightless [41].

Age-Related Decline and Resource Allocation: Age significantly influenced flight ability in both sexes, with peak performance observed at one day old. Flight capacity then declined with increasing age, with females experiencing a more rapid decrease (Figure 3 and Figure 4). The heightened flight ability of one-day-old males likely stems from their earlier emergence, allowing their spermatozoa to mature by the time females emerge. This necessitates a strong initial flight for locating mates [31,37,42,43]. Conversely, female ovaries mature after emergence, similar to other flighted moth species [11,37,44]. As females engage in mating and oviposition, their energy reserves shift towards reproduction, limiting the energy available for flight and leading to flight muscle deterioration. This likely explains the more rapid decline in flight ability observed in females compared to males. Additionally, *L. xylina* adults, similar to *L. dispar*, have vestigial mouthparts and do not feed after emergence. Their strongest flight coincides with the early stages when their bodily sugar and lipid reserves are highest. As they age, these reserves diminish, and the gradual deterioration of flight muscles hinders sustained flight [22,42]. Dissections revealed a gradual disappearance of fat bodies and flight muscles with increasing age, accompanied by hollow thoraces, bubble-like fat bodies, and filamentous flight muscles (unpublished data).

Flight Patterns and Comparison to Other Species: Our study identified complex flight patterns in *L. xylina* adults, including intermittent flight, sustained flight, and short-burst flight (Figure A1). These patterns resembled those observed in *Dendroctonus armandi* Tsai and Li adults [36]. The observed variations in flight number, distance, and duration suggest a complex interplay of factors influencing flight behavior.

Implications for Spread Potential: *L. xylina* is considered to be an invasive pest due to its broad host range and similarities with *L. d. asiatica* and *Lymantria dispar japonica* Motschulsky [29,45,46,47]. Our findings regarding the limited flight capability of females, especially after mating or during oviposition, suggest a potentially lower risk of long-distance dispersal compared to *L. d. asiatica*. Previous research reported average flight distances of 3.95 km to 7.50 km for unmated female *L. d. asiatica* (1 day old), measured over 8 h [22]. In our study, unmated female and male *L. xylina* moths (1 day old) had average flight distances of 3.975 km and 8.441 km, respectively, measured over 24 h. This pattern suggests that *L. xylina* adults may have a weaker flight capacity compared to *L. d. asiatica*, potentially reducing the risk of long-distance dispersal via ocean-going cargo ships.

Implications for Pest Management: The limitations of female flight ability, especially after copulation and during oviposition, suggest that pest management strategies targeting these stages could be effective in controlling the spread of *L. xylina*. For example, there are many management strategies for the female *L. dispar* moths before oviposition, e.g., light trapping and light suppression [48]. Some approaches, such as cleaning up surrounding branches and weeds, manual removal, and spraying of GPSO (Golden Pest Spray Oil), can be applied to destroy egg masses [49,50].

## 5. Conclusions

This study investigated the influence of sex, age, and flight duration on the flight ability of *L. xylina* adults. We found that females exhibited a significant decline in flight capability compared to males, likely due to their larger size and focus on reproduction. Their flight patterns primarily involved short distances, with reduced activity at night. Flight ability peaked for both sexes at one day old but declined rapidly thereafter, with females losing the ability to fly entirely by day five, suggesting a shift in energy allocation towards reproduction. Furthermore, extended flight duration negatively affected flight performance in both sexes, with females experiencing a more pronounced decline. Notably, the flight distances observed in this study were lower than those reported for *L. d. asiatica*, suggesting a potentially lower risk of long-distance dispersal for *L. xylina*, particularly for females. These findings on the limitations of female flight ability, especially after mating and during egg-laying, offer valuable insights for developing targeted pest management strategies to control the spread of *L. xylina*.

## Figures and Tables

**Figure 1 insects-15-00506-f001:**
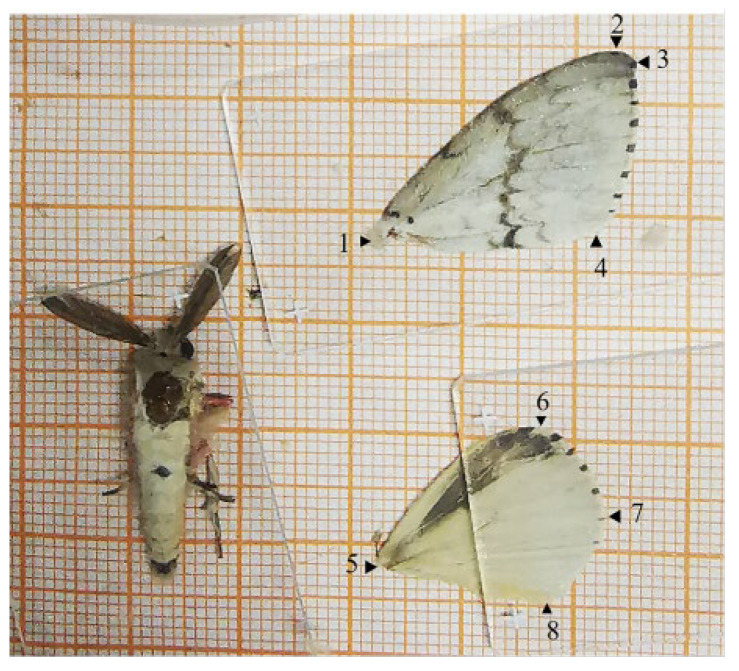
Measurement of the length and width of the wings of *L. xylina.* According to Shi et al. [23], the length of the forewing is the distance from the base of the costal vein (1) to the top of the apical angle, which is approximately the distance to the top (3) of the fourth radial branch (R_4_); the width of the forewing is the distance from the top (2) of the third radial branch (R_3_) to the edge of the anal angle (4); the length of the hindwing is the distance from the base of the costal vein (5) to the top (7) of the third medial branch (M_3_); and the width of the hindwing is the distance from the top (6) of the subcostal margin vein (S_c_) to the top (8) of the 3rd anal vein (A_3_).

**Figure 2 insects-15-00506-f002:**
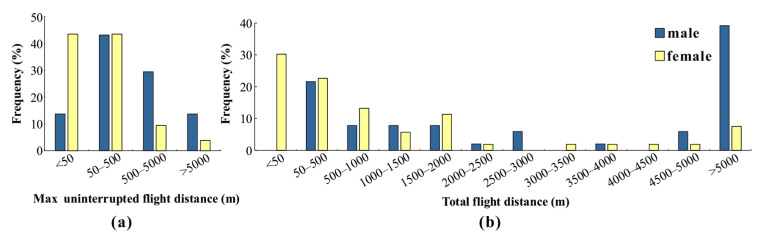
Frequency distribution of the (**a**) maximum uninterrupted flight distances and (**b**) total flight distances of *L. xylina* adults of different sexes.

**Figure 3 insects-15-00506-f003:**
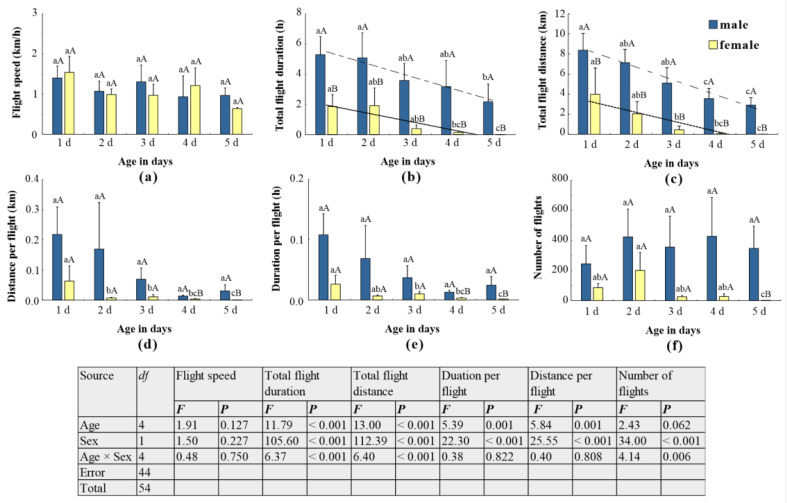
Flight activity of *L. xylina* adults of different sexes and ages: (**a**) flight speed, (**b**) total flight duration, (**c**) total flight distance, (**d**) distance per flight, (**e**) duration per flight, and (**f**) number of flights. The data in the figure are the mean ± SE; different lowercase letters represent significant differences in the flight parameters of adults of the same sex at different ages (*p* < 0.05, Duncan’s test); different capital letters represent significant differences in the flight parameters of male and female moths of the same age (*p* < 0.05, *t*-test).

**Figure 4 insects-15-00506-f004:**
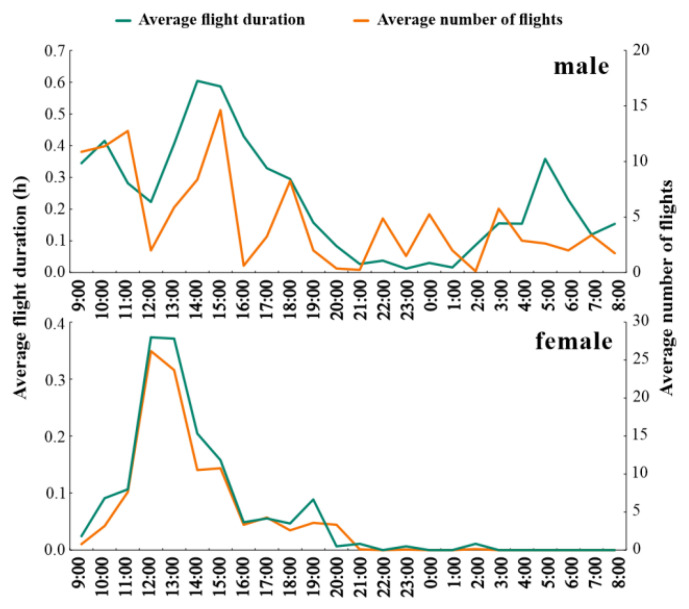
Flight regularity of 1-day-old male and female *L. xylina* moths on the flight mill.

**Figure 5 insects-15-00506-f005:**
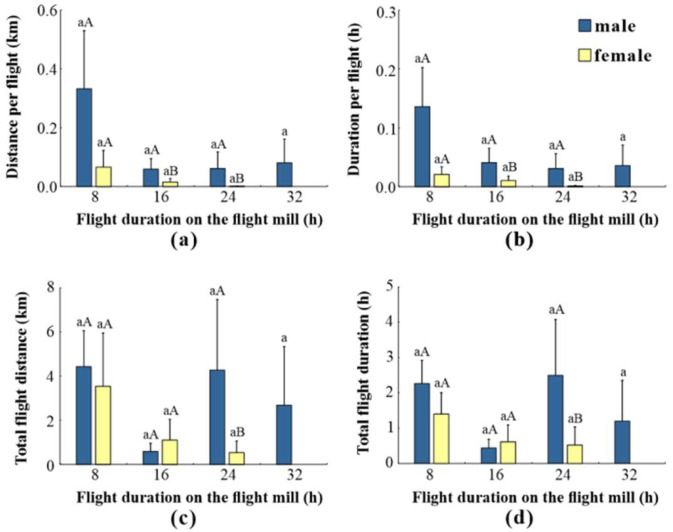
The flight ability of *L. xylina* adults at different flight durations on the flight mill: (**a**) distance per flight, (**b**) duration per flight, (**c**) total flight distance, and (**d**) total flight duration. The data in the figure are expressed as the mean ± SE; different lowercase letters represent significant differences in the flight parameters of adults of the same sex at the level (*p* < 0.05, Duncan’s test); different capital letters represent the flight parameters of female and male moths for the same flight duration at the level (*p* < 0.05, *t*-test).

**Figure 6 insects-15-00506-f006:**
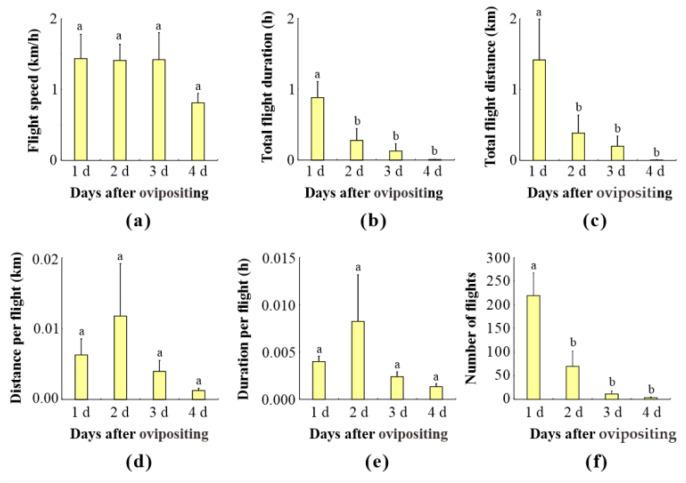
The flight parameters of *L. xylina* females at different days after oviposition: (**a**) flight speed, (**b**) total flight duration, (**c**) total flight distance, (**d**) distance per flight, (**e**) duration per flight, and (**f**) number of flights. Data in the figure are expressed as the mean ± SE; different lowercase letters indicate significant differences in the flight parameters of adults with different days of ovipositing at the level (*p* < 0.05, Duncan’s test).

**Table 1 insects-15-00506-t001:** List of morphological parameters of *L. xylina* adults.

	Details
Weight parameters	Preflight weight
Morphological parameters	Wing length
	Wing width
	Wing area (forewing area + hindwing area) × 2
	Abdominal length
	Abdominal width
	Body length
	Thorax width
Other parameters	Load ratio (body weight/wing area)
	Aspect ratio (wing length^2^/wing area)

**Table 2 insects-15-00506-t002:** The flight parameters of *L. xylina* adults with different mating statuses.

Sex	Mating Status	Flight Speed (km/h)	Total Flight Duration (h)	Duration per Flight (h)	Total Flight Distance (km)	Distance per Flight (km)	Number of Flights
Female	Mated	2.279 ± 0.414 a	0.378 ± 0.067 b	0.029 ± 0.011 a	0.914 ± 0.302 a	0.079 ± 0.042 a	17.0 ± 3.6 a
Female	Unmated	1.372 ± 0.097 a	1.601 ± 0.931 a	0.012 ± 0.002 a	1.962 ± 1.036 a	0.017 ± 0.005 b	151.0 ± 98.2 b
Male	Mated	1.380 ± 0.212 a	2.414 ± 0.909 a	0.045 ± 0.028 a	3.776 ± 1.446 a	0.075 ± 0.052 a	269.0 ± 201.7 a
Male	Unmated	2.177 ± 0.649 a	2.124 ± 0.792 a	0.071 ± 0.028 a	5.878 ± 2.890 a	0.187 ± 0.083 a	40.7 ± 9.0 b

Note: The data in the table are expressed as the mean ± SE, and different lowercase letters represent significant differences in the flight parameters of adults of the same sex with different mating status (*p* < 0.05, *t*-test).

**Table 3 insects-15-00506-t003:** The morphology and structural parameters of 1-day-old *L. xylina* adults with different total flight distances.

Sex	Total Flight Distance (m)	n	Weight (g)	Forewing Length (cm)	Thorax Width (cm)	Wing Area (cm^2^)	Forewing Aspect Ratio	Load Ratio (g/cm^2^)
Female	<500	3	1.29 ± 0.21 a	3.68 ± 0.09 a	0.76 ± 0.07 a	16.55 ± 1.72 a	2.80 ± 0.23 a	0.078 ± 0.008 a
Female	500–5000	11	0.75 ± 0.06 b	3.43 ± 0.12 a	0.67 ± 0.03 a	13.97 ± 0.75 a	2.91 ± 0.11 a	0.053 ± 0.003 b
Female	>5000	3	0.63 ± 0.02 b	3.32 ± 0.04 a	0.64 ± 0.04 a	12.75 ± 0.23 a	2.93 ± 0.02 a	0.050 ± 0.001 b
Male	<500	3	0.12 ± 0.02 a	2.24 ± 0.09 a	0.49 ± 0.08 a	8.41 ± 0.93 a	2.25 ± 0.15 a	0.015 ± 0.001 a
Male	500–5000	11	0.17 ± 0.02 a	2.54 ± 0.07 a	0.51 ± 0.02 a	9.00 ± 0.46 a	2.69 ± 0.09 a	0.019 ± 0.001 a
Male	>5000	8	0.20 ± 0.02 a	2.59 ± 0.05 a	0.55 ± 0.02 a	9.29 ± 0.56 a	2.70 ± 0.14 a	0.022 ± 0.002 a

Note: The data in the table are expressed as the mean ± SE; different lowercase letters represent significant differences in the parameters of adults of the same sex with different total flight distances at the level (*p* < 0.05, Duncan’s test).

## Data Availability

Data are contained within the article.

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
