# Peer review of "Evaluation of the Potential Flight Ability of the Casuarina Moth, Lymantria xylina (Lepidoptera: Erebidae)"

_insects, 2024, doi:10.3390/insects15070506_

Round 1

Reviewer 1 Report

Comments and Suggestions for Authors

The paper “Evaluation of the Potential Flight Ability of the Casuarina Moth, Lymantria xylina (Lepidoptera: Erebidae) is very complete in terms of laboratory tests and contains valuable information. The data and the information derived from this study can be used to study in-depth the flight ability and the dispersal capacity of the Casuarina Moth. However, the document requires some improvements in its current form. The first point that caught my attention is the lack of information regarding the economic losses or the problems that potentially can occasioned this organism. With regard to the methodology, the information is presented in a schematic format and occasionally there are instances where additional details are lacking, such as the abiotic conditions of the tests. Specific comments for questions and corrections are below:

Simple summary

L16-18. “…while female moths have weaker flight ability; female moths often rest on the branches and can sometimes fly short distances; and female moths could be attracted by the lights of ships and land on the hull or cargo to lay eggs.”  

Rewrite as:

“…while female moths have weaker flight ability. Female moths frequently rest on the branches and can sometimes fly short distances. Additionally, females may be attracted by the lights of ships and land on the hull or cargo to lay eggs.”

L23. Theoretical framework

Abstract

Review the verbal tenses e.g. Influenced, exhibited, declined, displayed

L25 Lymantria xylina Swinhoe (Lepidoptera: Erebidae)

L25 Lymantria dispar asiatica Vnukovskij and Lymantria dispar japonica Motschulsky (Lepidoptera: Erebidae)

L28-29 short distances / long distances, what are the mean distances?

Introduction

Some information could be added to the text: F. i. What is the importance of knowing about this pest species? Are there economic losses due to this pest? What is its potential economic/commercial impact?

L42 “Research on insect dispersal is very important for the controlling the spread of insects.

L45-46 “Primary methods for studying insect flight patterns include ovary dissection, field observation, mark-release-capture, and trapping [4-7].

L51 “…wing loading

L78 “…can disperse actively through the ballooning of early…”

L80 “…egg masses on substrates such as the surfaces…”

L81 “Female L. xylina moths exhibit a variety of behaviours…”

L86 “…of large number of females…”

L88 “…suggests the existence of long-distance flight capability.”

Materials and Methods

There are points than need more clarification: abiotic conditions of the experiments (T, HR and light), number of replicates…

L103 Sources of insects à Insects

L104-107 “Lymantria xylina pupae were collected from Pingtan County, Fujian Province of China (119°45′08.55″E, 25°25′13.28″N) in early June 2022, and placed individually in 200 mL transparent plastic boxes (25±1 °C, 40%-50% RH, and L16:D8). Adults were weighed after eclosion, and the sex, date, and serial number were recorded for future use.”

L116 Add the room temperature the relative humidity, and the illumination level (lux) of the different tests.

L141 Can you add the illumination level (lux) that the room where the tests were performed was at?

L145 How do you know what adults are capable to flight normally?

L145-146 “to anesthetize the adult individuals, 7 drops of 50% ethyl acetate were injected in a cotton within the insect box.”

L151-154 “To prepare for testing, the hair and scales on the pro-notum of the test insect were carefully removed. A small, homemade plastic tube (2.2 mm in diameter, 2.0 mm in height) was then attached to the pronotum using clear, quick-drying glue. The glue was allowed to set for a few seconds.”

-The number of the total number of individuals tested in each experiment is not clearly reported.

L160-164 “The insect flight information system was activated and the parameters were set, including a flight arm radius of 15 cm. The test insects naturally regained consciousness, and the flight mill automatically recorded the flight data. At the conclusion of the flight test, the test insects, along with the copper wire and small plastic tube, were removed. Subsequently, the copper wire and plastic tube were detached from the insects.”

Results

L213 “Potential Flight Ability of L. xylina Adults”

L262 “…males and females”

L333 (F, df, P)

Comments on the Quality of English Language

It would be advisable to submit the article to a linguistic review.

Reviewer 2 Report

Comments and Suggestions for Authors

I enjoyed reading this paper, in part because it shows that flight mills are still relevant.  The authors do a great job of using them to answer the question.  Overall, the paper is well presented and the data are thorough.  I have a few relatively minor changes to suggest.

In table one, does "chest width" mean thorax width? Morphological measurements of the wing are described in detail, but the authors need to include descriptions of all the measurements and how they were taken.

Table 3.  Why make categories of flight distance and compare those arbitrary groups?  Why not simply construct a graph of the effect of weight, etc. on flight distance?  Then apply linear regression or other statistical techniques. 

The form of the conclusion seems unusual. Why not write a single narrative in one paragraph, rather than a series of short statements with headings?  Most of the material is in the abstract, and is a summary of results, not conclusion related.

Line by line corrections

14 needs comma after Swinhoe

42 "for the controlling the spread" should read "for controlling the spread"

51 "wing flapping frequency" should be "wingbeat frequency"

52 needs comma before "etc."

213 and 259 L. xylina should not be italicized because the default font for subheadings is in italics. 

296 "Effects of Different Mating Statuses on Flight Ability" should read "Effects of Mating Status on Flight Ability"

377 "have a degenerated beak" should be "have vestigial mouthparts"

395 "This suggests" should read "This pattern suggests"

Comments on the Quality of English Language

There are minor suggestions which I have provided the authors above.

Round 2

Reviewer 1 Report

Comments and Suggestions for Authors

The paper, “Evaluation of the Potential Flight Ability of the Casuarina Moth, Lymantria xylina (Lepidoptera: Erebidae), has been reviewed satisfactorily, adressing all the questions raised by the reviewers. I would like to thank the authors and the other reviewers for their work improving the article. I believe that this article is ready to be accepted, but there is only one point that would be highly interesting to add if possible. The results obtained and their discussion lead to the conclusion that the flight limitations of the females, especially after copulation and during oviposition offer valuable insights for developing targeted pest management strategies to control the spread of this pest. This point is clear, but doubts may arise because the females are already ovipositing the offspring of the pest. Is there any other pest in which this has been observed and in which a control method has been developed based on this “behavior”? If such examples exist, it would be highly interesting to be discussed at the end of the discussion section. Otherwise, there is no need to delve into this topic further.
